# Polymer–Plasticizer Coatings for BTEX Detection Using Quartz Crystal Microbalance

**DOI:** 10.3390/s21165667

**Published:** 2021-08-23

**Authors:** Abhijeet Iyer, Veselinka Mitevska, Jonathan Samuelson, Scott Campbell, Venkat R. Bhethanabotla

**Affiliations:** Department of Chemical, Biological, and Materials Engineering, University of South Florida, Tampa, FL 33620-5350, USA; abhijeetiyer@usf.edu (A.I.); mitevskav@usf.edu (V.M.); jsamuels@usf.edu (J.S.); campbell@usf.edu (S.C.)

**Keywords:** plasticization, polymers, sensitivity, quantify, diffusion

## Abstract

Sensing films based on polymer–plasticizer coatings have been developed to detect volatile organic compounds (VOCs) in the atmosphere at low concentrations (ppm) using quartz crystal microbalances (QCMs). Of particular interest in this work are the VOCs benzene, ethylbenzene, and toluene which, along with xylene, are collectively referred to as BTEX. The combinations of four glassy polymers with five plasticizers were studied as prospective sensor films for this application, with PEMA-DINCH (5%) and PEMA-DIOA (5%) demonstrating optimal performance. This work shows how the sensitivity and selectivity of a glassy polymer film for BTEX detection can be altered by adding a precise amount and type of plasticizer. To quantify the film saturation dynamics and model the absorption of BTEX analyte molecules into the bulk of the sensing film, a diffusion study was performed in which the frequency–time curve obtained via QCM was correlated with gas-phase analyte composition and the infinite dilution partition coefficients of each constituent. The model was able to quantify the respective concentrations of each analyte from binary and ternary mixtures based on the difference in response time (*τ*) values using a single polymer–plasticizer film as opposed to the traditional approach of using a sensor array. This work presents a set of polymer–plasticizer coatings that can be used for detecting and quantifying the BTEX in air, and discusses the selection of an optimum film based on *τ*, infinite dilution partition coefficients, and stability over a period of time.

## 1. Introduction

Volatile organic compounds (VOCs) are chemicals that evaporate easily and become gases at ambient temperature and pressure. The Environmental Protection Agency (EPA) has made the monitoring of VOCs mandatory due to their health impacts, which range from headaches and nausea to cancer [1,2,3,4]. According to the Occupational Safety and Health Administration (OSHA), the permissible exposure limits (PEL) are 1 ppm for benzene, 200 ppm for toluene, and up to 100 ppm for xylene and ethylbenzene in air [5]. The current techniques used for VOC monitoring, such as photoionization, gas chromatography, mass spectroscopy, and e-nose sensors, are expensive, time-intensive, and demand rigorous sample preparation [6,7,8,9,10,11,12]. Chemical sensors based on piezoelectric transduction, such as quartz crystal microbalance (QCM) devices, show promising results in terms of linearity, stability, responsiveness, and selectivity [13,14,15].

Generally, the performance and sensitivity of rubbery polymers towards hydrocarbon sensing are superior to those of glassy polymers; however, they have low molecular selectivity due to their amorphous nature, the availability of large free volume elements, and their viscoelasticity [16]. Plasticization helps to modify the chemical and physical properties of a glassy polymer by decreasing the glass transition temperature, altering the pore dimensions, and increasing the free volume of the polymer film, thus enabling higher diffusion and sorption of the analyte molecules [17,18]. Sensing films based on polymer–plasticizer films have been developed in this study to detect volatile organic compounds in the air, specifically benzene, ethylbenzene, and toluene, using a QCM device. There has not been much work done using polymer–plasticizer coatings for BTEX detection in the gas phase using a QCM device, although some significant work was done in the liquid phase. Pejcic et al. [17,19,20] have previously shown that chemical sensors based on polymer-coated QCM sensors can detect and quantify BTEX compounds in aqueous solutions at lower concentrations on the order of a few ppm. The Josse group [16,21,22] reported the first-ever detection of benzene at ppb levels using SAW sensors with polymer–plasticizer coatings. Kaur et al. [18] from our laboratory studied the effect of plasticization on glassy polymer sorption using diisononyl cyclohexane-1,2-dicarboxylate (DINCH) and diisooctyl azelate (DIOA) as the plasticizers, and PEMA as the polymer.

## 2. Materials and Methods

For this work, the solvents benzene, ethylbenzene, and toluene were purchased from Sigma Aldrich at 99.9% purity. The homopolymers poly(ethyl methacrylate) (PEMA) with a molecular weight of 340,000 g/mol, poly(methyl methacrylate) (PMMA) with a molecular weight of 996,000 g/mol, polystyrene (PS) with a molecular weight of 280,000 g/mol, and a polystyrene/poly(methyl methacrylate) (PS/PMMA) block copolymer with a molecular weight of 300,000 g/mol were purchased from Sigma Aldrich. The plasticizers diisononyl cyclohexane-1,2-dicarboxylate (DINCH), diisooctyl azelate (DIOA), n-butyl stearate (BS), dibutyl sebacate (DBS), and di-n-butyl phthalate (DBP) were purchased from Scientific Polymer Products, Inc, Ontario, NY, USA.

Several factors were considered when choosing the type of plasticizers for BTEX sensing, including compatibility with the polymer, efficiency of the plasticization process, and the stability/permanence within the polymer over a long period of time [16]. The plasticizers used in this work were selected based on molecular weight, polarity, shape/spatial orientation, and the Hansen solubility parameter. Table 1 shows the chemical structures and properties of the plasticizers used while Table 2 shows the same for the polymers used in this work. The polymers used here are all glassy in nature and are chosen based on their Hansen solubility parameter values. To determine the compatibility of the plasticizer, Hansen solubility parameter values (Table 3) take into consideration various forces, such as van der Waals, hydrogen bonding, and polar interactions, with values close to those of the BTEX analytes indicating suitable candidates.

Previous work by Adapa et al. [23] showed that, amongst the DBS, DBP, and BS plasticizers, BS showed higher sorption and sensitivity in PEMA films owing to its linear structure and molecular weight. Therefore, in this work, experimentation was performed on four glassy polymers (PEMA, PMMA, PS, and PS/PMMA) and three plasticizers (DINCH, DIOA, and BS), the properties of which are shown in Table 3.

The vapor generation apparatus used in this study consisted of two QCM setups developed in our laboratory. Initially, the analysis of BTEX sorption in various polymer/plasticizer blends was performed at higher concentrations (a few 1000 ppm) using a vapor generation apparatus described in detail in our previous publication [24]. The analyte vapor streams were generated by bubbling nitrogen gas through impingers in this QCM setup. Sorption at lower concentrations (a few 100 ppm) was tested as well to assess the sensor’s ability to detect and differentiate BTEX constituents in the air. This apparatus has been described in detail in our previous work [25]. The analyte gas mixtures were generated using a syringe pump wherein the solvents were completely evaporated in the gas stream in the new modified QCM setup.

In each apparatus, a 5 MHz quartz crystal coated with a polymer–plasticizer film was exposed to the vapor of analytes of interest, such as benzene, toluene and ethylbenzene, with UHP nitrogen as the carrier gas, and left to equilibrate in order to calculate the weight fraction *w*_1_ of the solvent in the polymer, determined using the Sauerbrey [26] equation:(1)$$w_{1}=\frac{\Delta f}{\Delta f+\Delta f_{0}}$$
where Δ*f* refers to the frequency shift between that of the crystal coated in pure polymer and that of the crystal coated in polymer with sorbed solvent, and Δ*f*_0_ refers to the frequency change between that of the bare crystal and that of the crystal coated with pure polymer. All the sensor films were prepared by dissolving 0.5 g of the polymer–plasticizer blend (depending on the blend, plasticizer composition varied from 5 to 15%) in 20 mL of a suitable solvent (chloroform or toluene). The solutions were sonicated for an hour with heating to ensure homogeneity. A Laurell WS-400 B spin coater was then used to spin coat thin (~0.5 µm) films onto 5 MHz QCM crystals.

To model the diffusion of organic vapors in polymer systems, Masaro and Jhu [27] used kinetic studies of sorption and desorption to determine diffusion coefficients. The solution for a one-dimensional Fickian model (finite film) is:(2)$$\frac{M_{t}}{M_{equi}}=1-\frac{8}{\pi^{2}}\overset{\infty }{\underset{n=0}{\sum }}\frac{1}{{\left(2n+1\right)}^{2}}\exp\left[-\frac{Dt}{h^{2}}\pi^{2}{\left(2n+1\right)}^{2}\right]$$
where *M_t_* is the total mass of vapor absorbed by a film of thickness *h* at time *t*, *M_equi_* is the equilibrium sorption mass after infinite time, and *D* is the diffusion coefficient. Liu et al. [28] empirically developed an approximation of Equation (2) to bypass the infinite summation, as follows:(3)$$\frac{M_{i}}{M_{equi}}=1-\exp\left[-7.3{\left(\frac{Dt}{h^{2}}\right)}^{0.75}\right]$$

Equation (3) can be further simplified by defining *τ*, the response time constant, as $\tau =\frac{h^{2}}{14.161D}$, as follows:(4)$$\frac{M_{t}}{M_{equi}}=1-\exp\left[-{\left(\frac{t}{\tau }\right)}^{n}\right]$$

*M_t_* is proportional to *f*_0_ − *f* and *M_equi_* is proportional to *f*_0_ − *f_equi_*, wherein *f*_0_ is the purge frequency, *f* is the frequency at time *t*, and *f_equi_* is the equilibrium frequency. Therefore, Equation (4) can be rewritten in terms of frequency shifts, as follows:(5)$$\frac{f_{0-}f}{f_{0}-f_{equi}}=1-\exp\left[-{\left(\frac{t}{\tau }\right)}^{n}\right]$$

The frequency shifts were logged for BTEX constituents and fit to a rearrangement of Equation (5), as follows:(6)$$\Delta f(t)=\Delta f_{0}\left(1-e^{{\left(-t/\tau \right)}^{n}}\right)$$
where Δf(t) is the frequency shift as a function of time, $\Delta f_{0}$ is the equilibrium frequency shift due to sorption, τ is the response time constant, and *n* is a parameter chosen to provide the best fit of the model to the data (*n* = 1 was found to be optimal for all the films used here).

For multiple solvents, assuming solvents diffuse independently of each other, Equation (4) becomes:(7)$$M=\sum^{\;}M_{equi}\left[1-\exp\left[-\left(\frac{t}{\tau }\right)\right]\right]$$

For 2 solvents, Equation (7) was rewritten in terms of frequency shifts and gas-phase analyte compositions, as follows:(8)$$\Delta f(t)=\Delta f_{1}*\left(\frac{y_{1,amb}}{y_{1,pure}}\right)\left(1-e^{\left(-\frac{t}{\tau_{1}}\right)}\right)+\Delta f_{2}*\left(\frac{y_{2,amb}}{y_{2,pure}}\right)\left(1-e^{\left(-\frac{t}{\tau_{2}}\right)}\right)$$
where $\Delta f_{1}$ and $\Delta f_{2}$ are the equilibrium frequency shifts for analytes 1 and 2, τ_1_ and τ_2_ are the respective response time constants, and *y*_1,*pure*_ and *y*_2*,pure*_ are the vapor-phase mole fractions for the individual analyte runs. The vapor-phase mole fractions of analytes in a binary mixture *y*_1,*amb*_ and *y*_2,*amb*_ were then obtained by fitting the model to frequency–time data. Equation (8) can be extended to a ternary mixture by adding another term:(9)$$\Delta f(t)=\Delta f_{1}*\left(\frac{y_{1},amb}{y_{1},pure}\right)\left(1-e^{\left(-\frac{t}{\tau_{1}}\right)}\right)+\Delta f_{2}*\left(\frac{y_{2},amb}{y_{2},pure}\right)\left(1-e^{{\left(-\frac{t}{\tau_{2}}\right)}^{n}}\right)+\Delta f_{3}*\left(\frac{y_{3},amb}{y_{3},pure}\right)\left(1-e^{\left(-\frac{t}{\tau_{3}}\right)}\right)$$

Flory–Huggins theory, the preferred model for studying polymer–solvent thermodynamics, was applied to ternary systems in this study. The systems considered here consist of solvent (1), polymer (2), and plasticizer (3). We modified the Flory–Huggins theory for a ternary system consisting of a polymer, a plasticizer, and a solvent in a previous work [18]. Because the polymer/plasticizer ratio is fixed, these systems can be considered to be pseudo-binary systems, with appropriate definitions for the molar volume and molecular weight of the film, and the interaction parameter.

By introducing the polymer/plasticizer pseudo-component (*f*), the solvent activity can be written as:(10)$$lna_{1}=ln\emptyset_{1}+\left(1-\emptyset_{1}\right)\left(1-\left(\frac{V_{1}}{V_{f}}\right)\right)+\chi_{1f}{\left(1-\emptyset_{1}\right)}^{2}$$
where
(11)$$V_{f}=\frac{\alpha +1}{\frac{\alpha }{V_{3}}+\frac{1}{V_{2}}}$$
(12)$$\chi_{1f}=\chi_{13}\frac{\alpha }{\alpha +1}+\chi_{12}\frac{1}{\alpha +1}-\chi_{23}\frac{\alpha }{{\left(\alpha +1\right)}^{2}}$$
(13)$$\alpha =\frac{\emptyset_{3}}{\emptyset_{2}}$$

Values of $\chi_{1f}$ are obtained by fitting experimental data to Equation (10). The pseudo-binary Flory–Huggins model can be used to find the infinite dilution partition coefficients $\left(K^{\infty }\right)$ as follows:(14)$$K^{\infty }=\left[\frac{\rho_{f}\mathrm{RT}}{\Omega^{\infty }M_{1}P_{1}^{\mathrm{Sat}}}\right]$$
where $\rho_{f}$ is the density of the film, R is the gas constant, T is the temperature of the cell, *M*_1_ is the molecular weight of the solvent, P_1_^sat^ is the saturated vapor pressure of the solvent, and $\Omega^{\infty }$ is the weight-based infinite dilution activity coefficient, which is defined further as:(15)$$ln\Omega^{\infty }=ln\left(\frac{V_{1}}{V_{f}}*\frac{M_{f}}{M_{1}}\right)+1-\frac{V_{1}}{V_{f}}+\chi_{1f}$$
where *V*_1_, *V_f_*, *M*_1_, and *M_f_* are the volumes and molecular weights of the solvent and film, respectively, and $\chi_{1f}$ is the value obtained by fitting the experimental data to Equation (10). *M_f_*, which is the molecular weight of the film, can be written as:(16)$$M_{f}=\frac{V_{3}M_{2}+\alpha V_{2}M_{3}}{V_{3}+\alpha V_{2}}$$

The values of infinite dilution partition coefficients are given in Table 4 (below).

## 3. Results

The results can be categorized into three sub-parts. The sorption results demonstrate that a plasticized glassy polymer can be a better choice than the traditionally used rubbery polymer polyisobutylene (PIB), sensor response modeling shows how well our empirical model fits the experimental data and also how BTEX analytes can be differentiated and quantified from a mixture of contaminants at high as well as low concentrations, and, lastly, the selection of an optimum film is made based on sorption, *τ* ratios and stability results.

### 3.1. Sorption Results

The sorption of BTEX is more favorable in the plasticized glassy polymers tested here than in the traditionally used rubbery polymer, polyisobutylene (PIB). The activity vs. weight fraction curves shown in Figure 1a,b demonstrate that the sorption of benzene and toluene is larger in DINCH- and DIOA-plasticized PMMA films than in rubbery PIB films. Other plasticized glassy polymers showed similarly enhanced solubilities.

To assess the effects of the plasticizers, sensor measurements for the plasticized and unplasticized films were made. Figure 2a,b) shows the QCM response in the form of frequency–time curves for PMMA when exposed to toluene in the presence and absence of a plasticizer. The study shows that pure PMMA takes a large amount of time to equilibrate with toluene vapor compared to PMMA plasticized with 25% DINCH, which exhibited an immediate and pronounced frequency shift consistent with equilibration. This was also the case for PEMA, PS, and PMMA-PS, confirming that the response characteristics of a polymer-coated QCM sensor can be improved by adding a plasticizer. In this project, the performances of PEMA-, PMMA-, PS-, and PMMA-PS-coated QCM sensors with varying amounts of plasticizer were measured to optimize BTEX sensitivity.

Of the sensor films investigated in this work, PEMA plasticized with 5% DINCH or DIOA showed the maximum sorption and therefore the maximum response for all three analytes, benzene, toluene and ethylbenzene [25].

### 3.2. Sensor Response Model

Using the low-concentration QCM apparatus, the sensor response curves for the pure analytes benzene and toluene were collected and fit to Equation (6) to obtain the *τ* values and equilibrium frequency shifts for the identification of the analyte. For a PEMA-DIOA (10%) film, these values are shown in Table 5.

Binary mixtures of benzene and toluene were then analyzed using Equation (8) in conjnction with the *τ* values and equilibrium frequency shifts obtained from the single-analyte runs to obtain fitted values for the vapor-phase mole fractions *y*_1*,amb*_ and *y*_2*,amb*_. Table 6 shows the good agreement between the regressed and prepared mole fractions for a PEMA–DIOA (10%) polymer film. This result was consistent for several binary liquid mixture ratios—40–60, 60–40, 80–20 and 20–80 by volume for various analyte combinations, for which vapor mole fractions were calculated using an accurate phase equilibrium model—indicating that the binary model is applicable. Figure 3 shows the comparison between the experimental and estimated sensor response based on the model for an 80-20 benzene–toluene mixture.

This model was further applied to ternary mixtures of benzene, toluene, and ethylbenzene of various compositions (1–1–1, 1–1–3, and 1–3–1, respectively, by volume); it was found that the regressed vapor-phase mole fractions were in good agreement with the prepared compositions. Figure 4 shows the comparison between experimental and estimated sensor response based on the model for a 1–1–1 benzene–toluene–ethylbenzene mixture, and Table 7 shows the results for various other solvent ratios with a different film, PEMA–DINCH (5%). Again, the model used to detect the respective BTEX constituents and quantify their concentrations based on frequency shifts and *τ* values worked well. A large number of experiments involving exposing binary gas phase mixtures to various polymer–plasticizer films would have to be performed to determine an optimal film directly. Instead, we used simulations to determine the characteristics of films that would provide accurate values for gas phase mole fractions. Specifically, a number of data sets of Δ*f* versus t were created using Equation (8) by specifying *y*_1*,pure*_, *y*_2*,pure*_, *y*_1*,amb*_, *y*_2,*amb*_, Δ*f*_1_, Δ*f*_2_, *τ*_1_ and *τ*_2_. Random noise was then added to the frequency shift values. Several cases were examined including sets where *τ*_1_ and *τ*_2_ were close to each other while Δ*f*_1_ and Δ*f*_2_ were very different, runs where Δ*f*_1_ and Δ*f*_2_ were close to each other while *τ*_1_ and *τ*_2_ were very different, runs where both variables were close and runs where both variables were very different. The noisy data sets were then fit to Equation (8) to determine the values of *y*_1,*amb*_ and *y*_2,*amb*_ that provided the best fit to the data set, and these were compared to the values of *y*_1,*amb*_ and *y*_2,*amb*_ used to create the data set. Of course, adding no noise to the data sets would result in perfect agreement between the regressed values and those used to create the data set.

A binary set of solvents were taken from Sothivelr et al. [21] and *τ*_1_, *τ*_2_, Δ*f*_1_ and Δ*f*_2_ were set for each of the solvents, ethylbenzene and toluene, the concentrations being set as C_1_^v^ and C_2_^v^ for ethylbenzene and toluene, respectively. *τ*_1_ and *τ*_2_ are 204 and 76.7 s, Δ*f*_1_ = −10 Hz and Δ*f*_2_ = −6 Hz. The noise of the order of ±1 Hz is added to the data set, and Equation (5) is fit to the noisy data to determine C_1_^v^ and C_2_^v^, which gives the best fit to the data. For example, here in this work, three different sets of C_1_ and C_2_ values (analyte mole fractions expressed in concentrations of ethylbenzene and toluene in the gas mixture) C_1_ = 10 ppm and C_2_ = 10 ppm, C_1_ = 1 ppm and C_2_ = 10 ppm, and C_1_ = 10 ppm and C_2_ = 1 ppm were subdivided into four different cases, wherein the tau values, frequency shifts (Δ*f*), and both values *(*Δ*f* and *τ*) were kept apart to see how the predictions were affected.

It was found that the ability of the model to recover accurate gas phase mole fractions from noisy data was independent of whether Δ*f*_1_ and Δ*f*_2_ were similar or very different in values, as can be seen in Figure 5 below. However, the performance of the model deteriorated substantially when the *τ*_1_ value approached that of *τ*_2_, and when both Δ*f* and *τ* were kept close, as seen in Figure 5. We conclude that a candidate for the optimal film is one for which the ratio of *τ* values for analytes is large.

Using the two different apparatuses for generating vapor concentrations at high and low values, it was found that the time constant was independent of concentration from fits of the data using Equation (6). Thus, it can be inferred that, for a film of constant thickness, time constants obtained at higher concentrations will be applicable at lower ones as well.

### 3.3. Optimum Film

The apparatus developed for use at higher concentrations was used to screen potential polymer–plasticizer blends. Table 8 (below) gives the equilibrium frequency shifts and *τ* values for benzene, toluene, and ethylbenzene in various films along with their ratios. The desired percentages of plasticizers used in this work were chosen so that solubility enhancement would be observed without any viscoelastic effects. An optimum film would be expected to have *τ* values that are far apart, and therefore have high ratios. Of the twelve films studied, PS-BS (15%), PEMA–DINCH (5%), and PEMA–DIOA (5%) were the top three candidates based on *τ* ratios, and these were further investigated at lower concentrations. It was found that, for a film of constant thickness, *τ* was independent of the concentration between the orders of 100 ppm and 10,000 ppm.

As mentioned above, the three most promising films from Table 8 were selected based on the *τ* ratio, and new films of the same three materials were subsequently exposed to lower concentrations of the analyte vapor. However, the thicknesses of the films used here for higher and lower concentrations were different, hence the tau values for the same film are different in Table 8 and Table 9. On regressing the experimental data to the model in Equation (9), the *τ* values for the analytes in PEMA–DINCH (5%), PEMA–DIOA (5%) and PS–BS (15%) were found, and are given in Table 9.

### 3.4. Stability of the Plasticizer

The long-term monitoring of BTEX compounds in nature requires a sensor that responds linearly and reproducibly over a wide concentration range without degrading [28,29]. Plasticizer leaching is an important phenomenon to understand in order to predict the stability of sensor films over a long period. Leaching is dependent upon the size of the plasticizer molecule and the rate at which it diffuses through the polymer matrix [30]. Plasticizers with higher efficiency experience more rapid diffusion and, therefore, leach out more quickly. Polarity and hydrogen-bonding interactions between the polymer and the plasticizer also influence the permanence of the plasticizer in the matrix [22,31].

In this work, the stability of films was studied over a period of three months. The polymer–plasticizer films were spin-coated, resulting in thicknesses on the order of a few microns (0.2–1 micron). The frequency shifts for each analyte were studied once every month to determine whether the plasticizer was leaching out. The coated QCM crystal, when not used, was stored and exposed to air at room temperature.

As shown in Figure 6 and Figure 7 for a PS–BS (15%) film, the sorption frequency for each analyte over three months did not significantly change, indicating that no leaching of plasticizer occurred in any of the films over that period.

As mentioned previously, the choice of the optimum film is based on the sorption properties, *τ* ratios, and film stability. Although the *τ* ratios are important, higher sorption provides higher resolution, more rapid sensor response, and a better signal to noise ratio. Therefore, of the three films used here, PEMA–DINCH (5%) and PEMA–DIOA (5%) are superior to PS–BS (15%) by this criterion.

## 4. Conclusions

The incorporation of suitable plasticizers into a polymer-based sensor film can improve the selectivity for BTEX compounds by modifying its sorption properties, saturation dynamics, and stability over time. Poly(ethyl methacrylate) (PEMA), poly(methyl methacrylate) (PMMA), polystyrene (PS), and a PS/PMMA block copolymer were modified by introducing the plasticizers diisononyl cyclohexane-1,2-dicarboxylate (DINCH), diisooctyl azelate (DIOA), and n-butyl stearate (BS) in order to enhance BTEX sensitivity and selectivity. The relationship between plasticizer type and BTEX sensitivity is described in this contribution. It is shown that the sensitivity and performance of plasticized glassy polymers are superior to those of a commonly-used rubbery polymer, PIB, for this application, with PEMA–DINCH (5%) and PEMA–DIOA (5%) being optimal. The sensor films were able to detect, differentiate, and quantify BTEX constituents from binary and ternary mixtures to within experimental accuracy, making them good materials for BTEX detection in air.

## Figures and Tables

**Figure 1 sensors-21-05667-f001:**
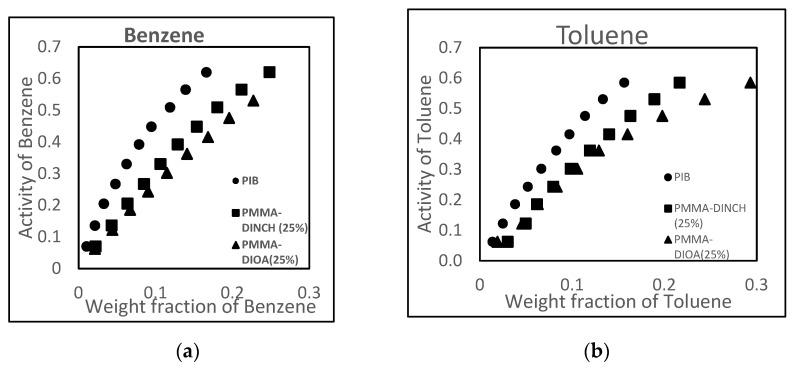
Comparison between sorption of benzene (**a**) and toluene (**b**) in a commonly used rubbery polymer, PIB, and two plasticized PMMA films.

**Figure 2 sensors-21-05667-f002:**
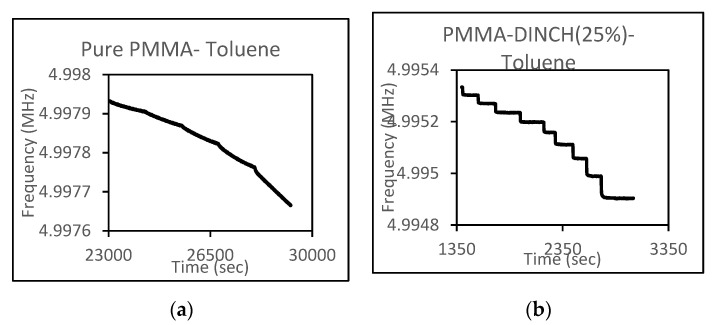
Frequency–time curves for pure PMMA-toluene (**a**) and a DINCH-plasticized PMMA system (**b**).

**Figure 3 sensors-21-05667-f003:**
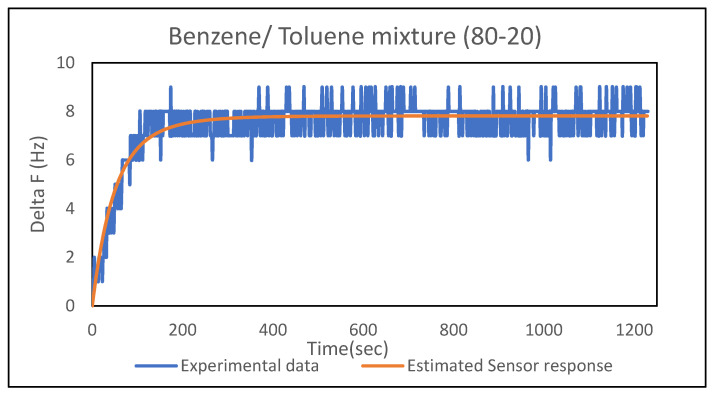
Comparison of experimental data and estimated sensor response for an 80–20 benzene–toluene binary mixture in a PEMA–DIOA (10%) film.

**Figure 4 sensors-21-05667-f004:**
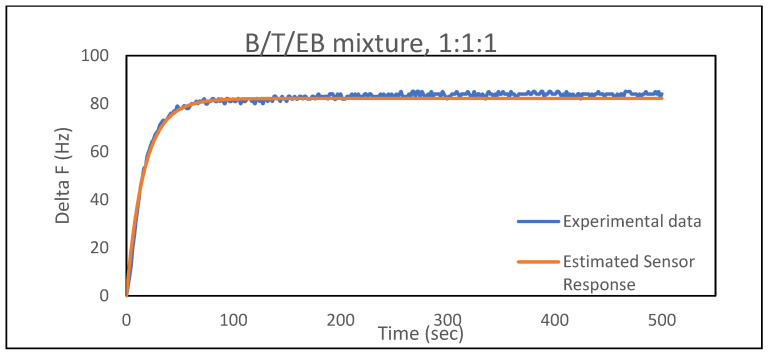
Comparison of experimental data and estimated sensor response from the model for B–T–EB mixture (1–1–1) in the PEMA–DINCH film.

**Figure 5 sensors-21-05667-f005:**
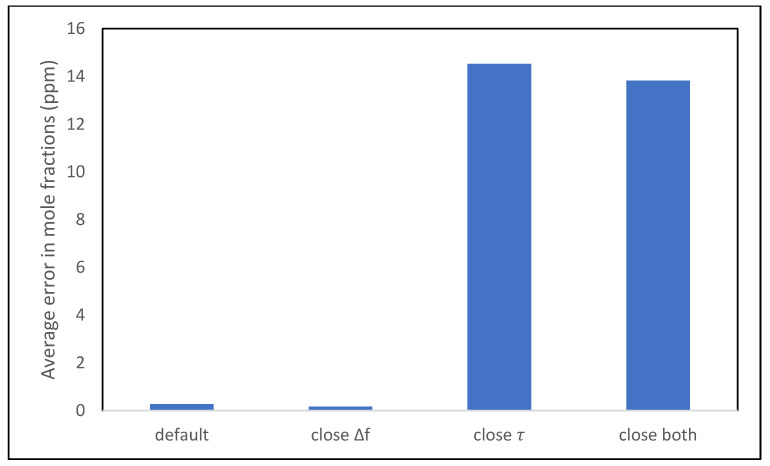
The average error in mole fraction values (ppm) for the four different cases.

**Figure 6 sensors-21-05667-f006:**
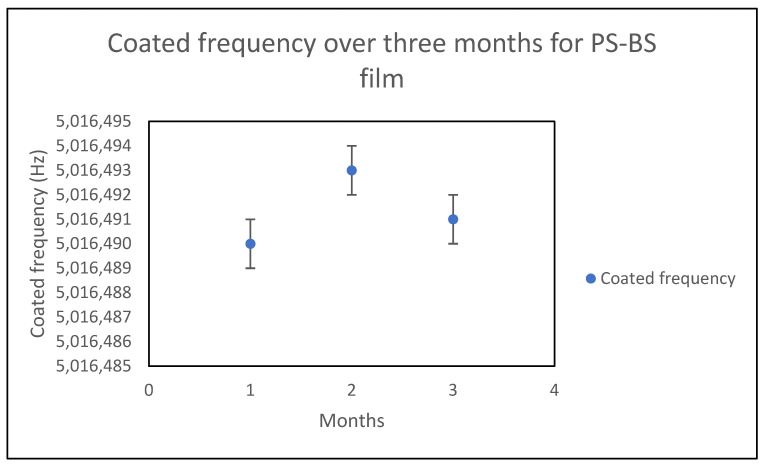
The shift in the coated frequency of the PS–BS (15%) film over three months.

**Figure 7 sensors-21-05667-f007:**
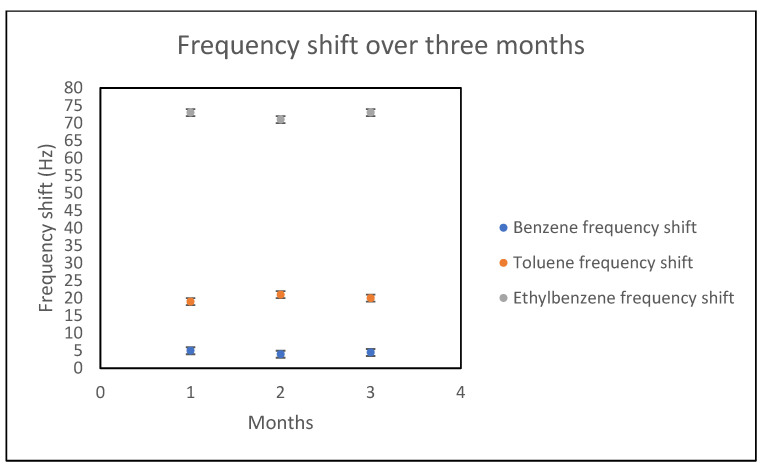
Frequency shifts for benzene, toluene and ethylbenzene in PS–BS (15%) film over three months.

**Table 1 sensors-21-05667-t001:** Structure and properties of plasticizers used.

Plasticizer	Structure	MW, g/mol	ρ, g/cm^3^
diisononyl cyclohexane-1,2-dicarboxylate (DINCH)	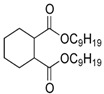	424.70	0.95
Diisooctyl azelate (DIOA)	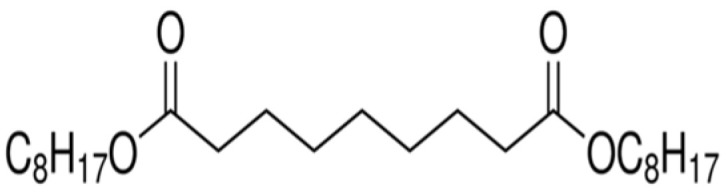	412.70	0.92
n-butyl stearate (BS)	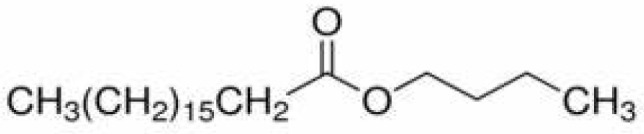	340.60	0.85

**Table 2 sensors-21-05667-t002:** Structure, molecular weight, and density of the polymers used.

Polymer	Structure	MW, g/mol	ρ, g/cm^3^
Poly(methyl methacrylate) (PMMA)	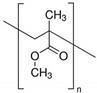	996,000	1.20
Poly(ethyl methacrylate) (PEMA)	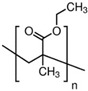	340,000	1.11
Polystyrene (PS)	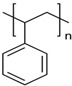	280,000	1.05
Poly(styrene-*block*-methyl methacrylate) (PS/ PMMA)	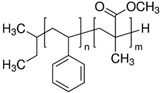	30,000	N/A

**Table 3 sensors-21-05667-t003:** Hansen solubility parameter and glass transition temperature of polymer and plasticizers used.

Polymer–Plasticizer-Solvent	Hansen Solubility Parameter (MPa^1/2^)	Glass Transition Temperature (°C)
Poly(methyl methacrylate) (PMMA)	18.6	105
Poly(ethyl methacrylate) (PEMA)	18.4	66
Polystyrene (PS)	18.6	100
Diisononyl cyclohexane-1,2-dicarboxylate (DINCH)	15.4	-
Diisooctyl azelate (DIOA)	16.7	-
n-butyl stearate (BS)	15.4	-
Benzene	18.5	-
Toluene	18.2	-
Ethylbenzene	17.9	-

**Table 4 sensors-21-05667-t004:** Values of infinite dilution partition coefficients ($K^{\infty }$) for benzene, toluene, and ethylbenzene in plasticized PEMA and PS films [25].

Solvent	Film	$K^{\infty }$
Benzene	PEMA-DINCH	766
Toluene	PEMA-DINCH	1908
Ethylbenzene	PEMA-DINCH	3899
Benzene	PEMA-DIOA	693
Toluene	PEMA-DIOA	1887
Ethylbenzene	PEMA-DIOA	4181
Benzene	PS-BS	318
Toluene	PS-BS	818
Ethylbenzene	PS-BS	1946

**Table 5 sensors-21-05667-t005:** Measured response times, *τ* (sec), and equilibrium frequency shifts for benzene and toluene in PEMA-DIOA (10%) film.

Analytes	Tau (*τ*), Sec	Frequency Shifts (Hz)
Benzene	41	6
Toluene	91	16

**Table 6 sensors-21-05667-t006:** Comparison of regressed mole fractions *y*_1*,amb*_ and *y*_2*,amb*_ to actual mole fractions *y*_1*,act*_ and *y*_2*,act*_ of binary mixtures of benzene and toluene at various concentrations.

Analyte 1	Analyte 2	Ratios	*y* _1*,amb*_	*y* _2*,amb*_	*y* _1,*act*_	*y* _2*,act*_
Benzene	Toluene	(20:80)	0.0001	0.0007	0.0002	0.0007
Benzene	Toluene	(50:50)	0.0006	0.0004	0.0005	0.0005
Benzene	Toluene	(60:40)	0.0007	0.0003	0.0007	0.0004
Benzene	Toluene	(80:20)	0.0009	0.0001	0.0009	0.0002
Benzene	Toluene	(40:60)	0.0004	0.0005	0.0004	0.0006

**Table 7 sensors-21-05667-t007:** Comparison of regressed mole fractions *y*_1,*amb*_, *y*_2,*amb*_, and *y*_3,*amb*_ from Equation (8) to actual mole fractions *y*_1,*act*_, *y*_2*,act*_ and *y*_3,*act*_ of benzene, toluene, and ethylbenzene, respectively.

Analyte 1	Analyte 2	Analyte 3	Ratios	*y* _1,*amb*_	*y* _2,*amb*_	*y* _3,*amb*_	*y* _1,*act*_	*y* _2,*act*_	*y* _3,*act*_
Benzene	Toluene	Ethylbenzene	(1:1:1)	0.003	0.0008	0.0002	0.003	0.0007	0.0002
Benzene	Toluene	Ethylbenzene	(1:1:3)	0.001	0.0008	0.0003	0.002	0.0005	0.0004
Benzene	Toluene	Ethylbenzene	(1:3:1)	0.001	0.002	0.0001	0.002	0.001	0.0001

**Table 8 sensors-21-05667-t008:** *τ* (s) and Δ*f_equil_* (Hz) for benzene, toluene, and ethylbenzene in various polymer–plasticizer coatings.

Polymer–Plasticizer	Benzene	Toluene	Ethylbenzene	*τ*_T_/*τ*_B_	*τ*_EB_/*τ*_T_	*τ*_EB_/*τ*_B_
PMMA–DINCH (15%)	*τ* = 193	*τ* = 304	*τ* = 303	1.57	1.00	1.57
	Δ*f* = 65	Δ*f*=55	Δ*f* = 55			
PMMA–DIOA (15%)	*τ* = 250	*τ* = 257	*τ* = 333	1.03	1.30	1.33
	Δ*f* = 146	Δ*f* = 127	Δ*f* = 119			
PMMA–BS (15%)	*τ* = 65	*τ* = 60	*τ* = 90	0.93	1.49	1.38
	Δ*f* = 221	Δ*f* = 207	Δ*f* = 200			
PMMA/PS–DINCH (10%)	*τ* = 248	*τ* = 738	*τ* = 712	2.97	0.97	2.87
	Δ*f* = 29	Δ*f* = 26	Δ*f* = 30			
PMMA/PS–DIOA (15%)	*τ* = 156	*τ* = 169	*τ* = 228	1.09	1.35	1.47
	Δ*f* = 166	Δ*f* = 147	Δ*f* = 141			
PMMA/PS–BS (10%)	*τ* = 104	*τ* = 109	*τ* = 137	1.05	1.25	1.32
	Δ*f* = 128	Δ*f* = 125	Δ*f* = 116			
PEMA–DINCH (5%)	*τ* = 25	*τ* = 40	*τ* = 63	1.57	1.59	2.49
	Δ*f* = 102	Δ*f* = 89	Δ*f* = 79			
PEMA–DIOA (5%)	*τ* = 204	*τ* = 281	*τ* = 426	1.37	1.52	2.09
	Δ*f* = 145	Δ*f* = 129	Δ*f* = 120			
PEMA–BS (5%)	*τ* = 47	*τ* = 77	*τ* = 88	1.65	1.14	1.89
	Δ*f* = 64	Δ*f* = 63	Δ*f* = 55			
PS–DINCH (15%)	*τ* = 170	*τ* = 299	*τ* = 372	1.76	1.24	2.19
	Δ*f* = 60	Δ*f* = 51	Δ*f* = 52			
PS–DIOA (15%)	*τ* = 94	*τ* = 286	*τ* = 311	3.03	1.09	3.29
	Δ*f* = 37	Δ*f* = 33	Δ*f* = 33			
PS–BS (15%)	*τ* = 173	*τ* = 243	*τ* = 586	1.40	2.41	3.37
	Δ*f* = 42	Δ*f* = 33	Δ*f* = 37			

**Table 9 sensors-21-05667-t009:** *τ* values for benzene, toluene, and ethylbenzene for the three most promising polymer–plasticizer coatings.

Polymer–Plasticizer	Benzene	Toluene	Ethylbenzene	*τ*_T_/*τ*_B_	*τ*_EB_/*τ*_T_	*τ*_EB_/*τ*_B_
PEMA–DINCH (5%)	*τ* = 155	*τ* = 240	*τ* = 640	1.55	2.67	4.13
PEMA–DIOA (5%)	*τ* =220	*τ* =380	*τ* = 620	1.73	1.63	2.82
PS–BS (15%)	*τ* = 50	*τ* = 107	*τ* = 450	2.14	4.21	9

## Data Availability

Not applicable.

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
