# Peer review of "Polymer–Plasticizer Coatings for BTEX Detection Using Quartz Crystal Microbalance"

_sensors, 2021, doi:10.3390/s21165667_

Round 1

Reviewer 1 Report

Please disclose (in the text) the relationship between the data presented in this manuscript and the reference  [24] -  "submitted" is not informative and should not be listed.

Author Response

Comment- Please disclose (in the text) the relationship between the data presented in this manuscript and the reference  [24] -  "submitted" is not informative and should not be listed.

Response- The only overlap between reference [24] and this paper is the values reported in Table 4, which were extracted from the reference [24].

Reviewer 2 Report

I have read the paper several times and it contains for me adequate novelty; moreover the scientific methods and strategy are thoroughly described. The approach offered by the authors represents a real and powerful and efficient alternative to the sometimes elaborate application of sensor arrays.

I appreciate not only the use of plasticizers to improve the performances of glassy polymers, but above all the following Sensors Response Modeling.

In my opinion the manuscript is practically in a status suitable for publication. I just wish to give some minor hints that should be useful for further publications and developments on the same subject:

  • even according to the authors' statement, benzene, ethylbenzene, toluene and xylenes are all referred as BTEX: therefore it should be appropriate to consider also xylene (it should be done in a future work).
  • the stability of plasticizer has been dealt with; nevertheless chemical sensors are also wholly characterized by other parameters such as reproducibility, recovery time, costs, ease of fabrication, etc. that should be concerned with.
  • also a complete microscopy, morphological and structural analysis should be useful for a complete comprehension and rationale of the sensor behaviour and should be the topic of a continuation of this research.
  • I suggest to cite these papers 10.1016/j.snb.2013.02.057 among references [6-11]

Author Response

Comments- 

  1. even according to the authors' statement, benzene, ethylbenzene, toluene and xylenes are all referred as BTEX: therefore it should be appropriate to consider also xylene (it should be done in a future work).
  2. the stability of plasticizer has been dealt with; nevertheless chemical sensors are also wholly characterized by other parameters such as reproducibility, recovery time, costs, ease of fabrication, etc. that should be concerned with.
  3. also a complete microscopy, morphological and structural analysis should be useful for a complete comprehension and rationale of the sensor behaviour and should be the topic of a continuation of this research.
  4. I suggest to cite these papers 10.1016/j.snb.2013.02.057 among references [6-11]

Response: 

Thank you for the suggestions for future publications/work.

  1. Yes, we are planning to include xylene in future work.
  2. We will take these parameters into consideration for future work.
  3. Adding a microscopy component will be a useful topic of research.
  4. It has been added.

Reviewer 3 Report

Iyer et al. used quartz crystal microbalances (QCMs) to detect the absorption of volatile organic compounds (VOCs)-BTEX at the sensing films based on polymer-plasticizer coatings. In this manuscript, they reported that how the sensitivity and selectivity of a glassy polymer film for BTEX detection can be altered by adding a precise amount and type of plasticizer. The film saturation dynamics and model of BTEX absorption into the bulk of the sensing film are also presented in this work. This study demonstrates that the sensitivity and performance of plasticized glassy polymers are superior to those of a commonly-used rubbery polymer and shows how well the empirical model fits experimental data. The sensor films are able to detect, differentiate, and quantify BTEX constituents from binary and ternary mixtures to within experimental accuracy making them good materials for BTEX detection in air. This is quite interesting. The manuscript is well written and all the explanations are reasonable. I recommend publication of this manuscript after some minor revisions.

  1. It is suggested that the authors should provide the more original data graphs of the frequency-time curve obtained by QCM in the section 3.1, section 3.2 and section 3.3.
  2. There are too many tables in the manuscript. Some tables can be replaced with graphs. Some tables are suggested to place in the supplementary information.
  3. It is suggested a more description on the QCM experiments.
  4. The figures are suggested to be modified. The size of the data points in two panels of Figures is inconsistent.
  5. Additional explanation is suggested regarding the difference between the calculated τ values in Table 9 and experimental τ values in Table 8.

Author Response

Comments: 

  1. It is suggested that the authors should provide the more original data graphs of the frequency-time curve obtained by QCM in the section 3.1, section 3.2 and section 3.3.
  2. There are too many tables in the manuscript. Some tables can be replaced with graphs. Some tables are suggested to place in the supplementary information.
  3. It is suggested a more description on the QCM experiments.
  4. The figures are suggested to be modified. The size of the data points in two panels of Figures is inconsistent.
  5. Additional explanation is suggested regarding the difference between the calculated τ values in Table 9 and experimental τ values in Table 8.

Response:

  1. The experimental output is in excel format and we plotted a graph using the data. These are graphs of original data.
  2. In our opinion, the only table that can be conveniently replaced by a graph is Table 4 and the graph would take up more space than the table does, so we prefer to leave it as it is.
  3. Since the descriptions of the QCM experiments and apparatus were long, we have referred to our previous publications.
  4. It has been fixed.
  5. We have mentioned that for a film of constant thickness, tau is found to be independent of concentration, however, the films used at lower and higher concentrations are of different thicknesses, hence the difference in tau values in table 8 and 9. We have added that statement in the Optimum Film section.

Reviewer 4 Report

It is a nice work that, however, still must be improved.

Please, define abbreviations when first used (e.g. EPA, OSHA).

There is an error in the unity for temperature.

What are the heating temperature and concentrations of solutions ?

There is an error in "proportionate".

In "The noisy data sets were then fit to equation 8 to determine the values of y1, amb 322 and y2, amb that provided the best fit to the data set and these were compared to the values 323 of y1, amb and y2, amb used to create the data set. Of course, adding no noise to the data sets 324 would result in perfect agreement between the regressed values and those used to create 325 the data set.", please, provide more information on these results.

In "The apparatus developed for use at higher concentrations was used to screen 349 potential polymer-plasticizer blends.", please, provide more details on the differences of each apparatus in a proper section.

In the conclusions, it is hard to say that performance is better than rubbery polymers when just one compound was tested from this group.

Author Response

Comments:

1. Please, define abbreviations when first used (e.g. EPA, OSHA).

2. There is an error in the unity for temperature.

3. What are the heating temperature and concentrations of solutions ?

4. There is an error in "proportionate".

5. In "The noisy data sets were then fit to equation 8 to determine the values of y1, amb 322 and y2, amb that provided the best fit to the data set and these were compared to the values 323 of y1, amb and y2, amb used to create the data set. Of course, adding no noise to the data sets 324 would result in perfect agreement between the regressed values and those used to create 325 the data set.", please, provide more information on these results.

6. In "The apparatus developed for use at higher concentrations was used to screen 349 potential polymer-plasticizer blends.", please, provide more details on the differences of each apparatus in a proper section.

7. In the conclusions, it is hard to say that performance is better than rubbery polymers when just one compound was tested from this group.

Response:

  1. Abbreviations for EPA, OSHA have been added.
  2. Glass transition temperatures are in °C. 
  3. The blend of polymer and plasticizer is prepared gravimetrically by adding the desired amount of plasticizer to 0.5 g of the polymer, which is dissolved in chloroform solvent (20 ml), sonicated for an hour, and left overnight before being spin-coated onto the QCM. All these steps are carried over at room temperature.
  4. It has been fixed.
  5. We have addressed the concern in section 3.2 by adding figure 4 and text explaining it.
  6. We have added a couple of sentences stating the differences in each apparatus in section 2. The complete details have been cited in our previous publications.
  7. In all instances in the paper except one, we talked about only PIB, and in the one place we didn’t, we have changed text.